# Motor variability during resistance training: Acceleration signal as intensity indicator

**Miguel López-Fernández, Fernando García-Aguilar**  *, **Pablo Asencio, Carla Caballero, Francisco J. Moreno**  , **Rafael Sabido**

Sport Sciences Department, Laboratory of Motor Control and Learning, Miguel Hernández University of Elche, Elche, Alicante, Spain

* fernando.garciaa@umh.es

**Data Availability Statement:** The minimal anonymized dataset necessary to replicate the study findings has been deposited in Zenodo. The

## Abstract

Analysis of variability in physiological time series has been shown to be an indicator of the state of the organism. Although there is evidence of the usefulness of analysis of the amount and/or structure of variability (complexity) in cycling actions, there is limited knowledge about its application in resistance exercise. The aim of this study is to find out whether variability in acceleration signals can be an indicator of intensity level in a squat task. For this purpose, an experimental design was developed in which the following participated seventy-two participants (age = 25.7 ± 4.4 years; height = 169.2 ± 9.8 cm; body mass = 67.7 ± 11.2 kg; ratio 1RM/body mass = 1.4 ± 0.3). They performed four repetitions of back squat at loads of 10%, 30%, 50%, 70%, and 90% of 1RM. Acceleration during the exercise was recorded using an inertial measurement unit (IMU) and a force platform. The variability of the movement was then analyzed using Standard Deviation (SD), Detrended Fluctuation Analysis (DFA), Fuzzy Entropy (FuzzyEn), and Sample Entropy (SampEn). For the IMU and for the force platform, significant effects were observed in all variables (p < 0.001). In pairwise comparisons, IMU showed a significant increase in motor complexity with increasing intensity, among most intensities, in DFA, FuzzyEn and SampEn. Differences in force platform were more limited, and only DFA detected differences between most intensities. The results suggest that measures of signal and acceleration variability may be a useful indicator of the relative intensity at which a squat exercise is performed.

## Introduction

Motor variability is a characteristic of human movement defined by the normal variations that occur in motor performance across repetitions during the execution of a motor skill [1]. This variability indicates that each repetition has different motor patterns from the next repetition so that one repetition will never be the same as the next regardless of a person's level of experience [2]. Motor variability can be considered an adaptive phenomenon, as a consistent outcome can be achieved through the execution of different motor patterns [3]. Furthermore, variability is also considered a phenomenon that aids motor learning through the exploration of possible solutions in a motor skill [4].

data can be accessed freely at the following DOI: https://doi.org/10.5281/zenodo.12786141.

**Funding:** This manuscript is related to a national project funded by the Ministerio de Ciencia e Innovación [PID2019-109632RB-I00] by Rafael Sabido and Francisco J. Moreno. The contribution of Miguel López-Fernández was funded by the Generalitat Valenciana, Spain [grant number: CIACIF/2021/452]. The contribution of Fernando García-Aguilar was funded by the Generalitat Valenciana, Spain [grant number: ACIF/2021/159]. The contribution of Pablo Asencio has been funded by the Ministerio de Ciencia e Innovación [PRE2020-091858]. The funders had no role in the study design, data collection and analysis, decision to publish, or preparation of the manuscript.

**Competing interests:** The authors have declared that no competing interests exist

There are different procedures for analyzing motor variability, and specifically, the analysis of variability structure has been related to self-organization processes, providing important information about the dynamics and adaptive capacity of living organisms [1, 5]. The analysis of variability structure can be performed through different procedures; among them, the use of non-linear tools allows us to better study motor behavior variations and how they emerge over time [1], providing additional information on the dynamics of variability [6] and its complexity [7]. Nonlinear tools are used to assess movement complexity or variability movement, each analyzing different properties of variability [6]. Among them, the DFA [8] is a fractal measure that analyzes the long-range autocorrelation of the signal to evaluate complexity. Another tool used to assess complexity is FuzzyEn [9], which assesses the degree of signal irregularity, with higher entropy values indicating higher irregularity.

A constraint refers to any factor that limits or influences motor behavior, and these constraints can be of an internal (flexibility, muscular strength, etc.) or external nature (weight of an object, specific task rules, etc.) [3, 10] In strength training, there are different constraints that are commonly used, such as the number of repetitions and sets to be performed, execution velocity, rest periods, or the load to be lifted, among others. Several studies have manipulated different constraints in strength training to analyze their influence on the variability structure, such as using tasks that induce fatigue [11], modifying the type of contraction [12], using unstable platforms [13], or sporting equipment like a rugby ball [14].

Load intensity is one of the most commonly used variables to modify the intensity of resistance training [15], but to the author's knowledge there are few studies analyzing the influence of some load intensities on the variability structure [16]. Pethick et al. [16] analyzed the complexity of force production in the knee extensors isometrically to see how this motor complexity is modified in the presence of fatigue. They found that fatigue reduced the complexity of force production by limiting the adaptability of the neuromuscular system to external constraints. However, it should be noted that the aim of this study was to analyze the complexity of force production during fatiguing tasks.

Examining force variability concerning resistance training load intensity can deepen insights into motor adaptation and the dynamics of the motor system response to training conditions. The application of non-linear tools further refines the assessment of force variability complexity. Given the increasing interest in employing non-linear tools for movement complexity analysis in resistance training [17] and recognizing that constraints like load intensity may alter movement complexity, this study aims to investigate the impact of a constraint, specifically load intensity, on the variability structure during a squat exercise.

## Methods

### Participants

Seventy-two healthy participants (age = 25.7 ± 4.4 years; height = 169.2 ± 9.8 cm; body mass = 67.7 ± 11.2 kg; one repetitions maximum (1RM) in squat = 95.5 ± 28.7 kg; ratio 1RM/body mass = 1.4 ± 0.3) took part in the study (Table 1). Recruitment of participants began on March 1, 2021, and ended on June 31, 2021. Prior to participation, each subject provided written informed consent, which was approved by the ethics committee of the University

**Table 1. Participant characteristics: Gender, n, age, height, body mass, 1RM squat, 1RM/body mass.**

| Gender | N | Age | Height | Body mass | 1RM squat | 1RM/Body mass |
|--------|-----|------|--------|-----------|-----------|---------------|
| Male | 37 | 25.9 | 175.1 | 73.5 | 112.9 | 1.5 |
| Female | 35 | 35.4 | 163.2 | 61.6 | 77.1 | 1.3 |

(PID2019-109632RB-100), and which adhered to the Declaration of Helsinki. The Office of Responsible Research (OIR), as the University's ethics committee, approves the research ethics statement under register 2019.417.E.OIR; 2020.34.E.OIR and reference DCD.RSS.02.19.

Participants were instructed to maintain their normal lifestyle, including nutritional and hydration states. Caffeine intake was not allowed in the 3 h before measurements. In addition, resistance training sessions were not allowed in the 72 h before the experimental sessions. All participants had at least one year of experience in resistance training in the squat exercise and they had no injuries during the six months *prior to the study*.

## Procedure

Participants attended three testing sessions separated by at least 72 h. To avoid experimental data bias variability, participants were scheduled at the same time for each session. On day one the participants performed the 1RM squat test after being familiarized with the warm-up protocol. The squat depth was in the parallel squat position [18]. The 1RM test was performed following an established protocol, which required the load to be progressively increased until the mean propulsive velocity was below 0.5 m/s as several studies advocate the use of movement velocity for 1RM estimation [19]. The estimated 1RM was automatically calculated by the specialized software of the linear position transducer (T-Force System, Ergotech, Spain). In the second and third sessions the participants performed the experimental protocol, which consisted of a total of five sets of four consecutive repetitions (e.g., no rest between them) in the squat exercise. The back-squat exercise required the subjects to rest the bar on their trapezius and then squat to the parallel position, which was defined as when the greater trochanter of the femur was lowered to the same level as the knee. Adequate depth of the repetition was ensured by placing a small bar at the individual height at which each participant had to descend to touch it lightly with the buttocks and perform the ascent. Participants performed one set with each of the following loads: 10%, 30%, 50%, 70%, and 90% of 1RM at a preferred velocity. The order of loads was balanced between the participants. Rest intervals between sets were 4 min.

## Data collection and analysis

To determine RM and monitor mean propulsive velocity during the set protocol, a linear encoder system was employed (T-Force Dynamic Measurement System, Ergotech, Murcia, Spain). The platform force data required for analysis were obtained using a Kistler (Switzerland, Mode 9287BA), which was calibrated with InstaCal software (Measurement Computing Corporation, Norton, USA) before starting the protocol. Acceleration signals were recorded utilizing one IMU made by STT-System (San Sebastián, España). The IMU was situated in the lumbar region at the level of the iliac crest between the L3 and L5 vertebra. Both the force platform and IMU were registered at 100 Hz through the iSenTM software (STT-System, San Sebastián, Spain). The SD and variability structure of the acceleration modulus time series were analyzed. The modulus (or vector) of the acceleration was obtained by calculating the square root of the sum of the values recorded on the Z, X and Y axes squared. The series were analyzed using DFA for the fractal scaling, FuzzyEn and SampEn, for analyzed the regularity or predictability. FuzzyEn was calculated using a protocol set out by Chen et al. [9], the parameters used were m = 2, r = 0.2 * SD, and n = 2. SampEn was calculated using a protocol set out by Yentes et al. [20], the parameters used were m = 2 and r = 0.2 * SD. DFA was calculated according to Peng et al. [8] and using windows of one-second duration. The windows' duration was adjusted based on the sampling frequency. Therefore, we used 4 initial windows and 100 final windows. The use of IMUs for the study of variability through acceleration in such

protocols has shown moderate to good reliability and a relatively low standard error of measurement [21].

## Statistical analysis

The data obtained were analyzed using JASP 0.17.1.0 (Jeffreys's Amazing Statistics Program). The normality of the variables was assessed using the Kolmogorov-Smirnov test. Repeated measures ANOVAs with one factor (5 x 1) were then performed on the different loadings, using analyses to see if there were differences in the different percentage loadings on the variables' DFA, FuzzyEn, SampEn, and SD. A post-hoc analysis was performed to analyze pairwise comparisons and the Bonferroni correction was applied. The level of statistical significance was considered $p < 0.05$. Partial eta squared ($\eta^2 p$) was calculated as a measure of effect size and to provide a proportion of the total variance attributable to the factor. Effect size values $\geq 0.64$ were considered strong, around 0.25 were considered moderate, and $\leq 0.04$ were considered small [22].

## Results

ANOVA analysis showed overall significant effect between loads on the IMU device for the DFA (F = 94.697; $p < 0.001$; n2p = 0.575), FuzzyEn (F = 182.526; $p < 0.001$; n2p = 0.720), SampEn (F = 75.940; $p < 0.001$; n2p = 0.517) and SD (F = 32.497; $p < 0.001$; n2p = 0.314) measures. For pairwise comparisons, significant differences were found between all the loads except for the comparison between [10%-30%] on the DFA and SampEn measures. For FuzzyEn there were statistically significant effect between all loads. In the case SD were obtained significant differences between all loadings except for the comparisons between [10%-30%], [10%-50%], [30%-50%] where there were no significant differences. Fig 1 shows the significant differences between each of the loadings in the IMU.

For the force platform, significant effect were observed between loadings for the DFA (F = 95.062; $p < 0.001$; n2p = 0.569), FuzzyEn (F = 20.920; $p < 0.001$; n2p = 0.225), SampEn (F = 88.830; $p < 0.001$; n2p = 0.552), and SD (F = 74.912; $p < 0.001$; n2p = 0.510) measures. In pairwise comparisons, significant differences were found between all loadings except for comparisons between [10–30%] and [30–50%] on the DFA measure. Significant differences were only found for FuzzyEn when comparing the 10% loadings with the rest, while no significant differences were found for the other loadings. Regarding SampEn and SD, significant differences were found between all loads except for the [50–70%], [50–90%], and [70–90%] comparisons where there were no differences. Fig 2 shows the significant differences between each of the loads on the force platform.

## Discussion

The main aim of this experiment was to analyze the influence of a specific constraint, namely load intensity, on the variability structure during the squat exercise. The experiment sought to understand how variability, in relation to the resistance training load, affects the motor adaptation and dynamic nature of the motor system's response during the squat. This allows us to understand whether variability is sensitive to changes in the intensity of the load in a strength exercise. The results from this study indicate that load intensity constrains participants' movement and increases movement complexity during the execution of the squat. The IMU recordings tend to show a decrease in the linear measure of variability (SD) and an increase in motor complexity (entropy increase and a reduction of DFA) with an increase of the load due to movement adaptations. On the other hand, our results from the force platform showed that

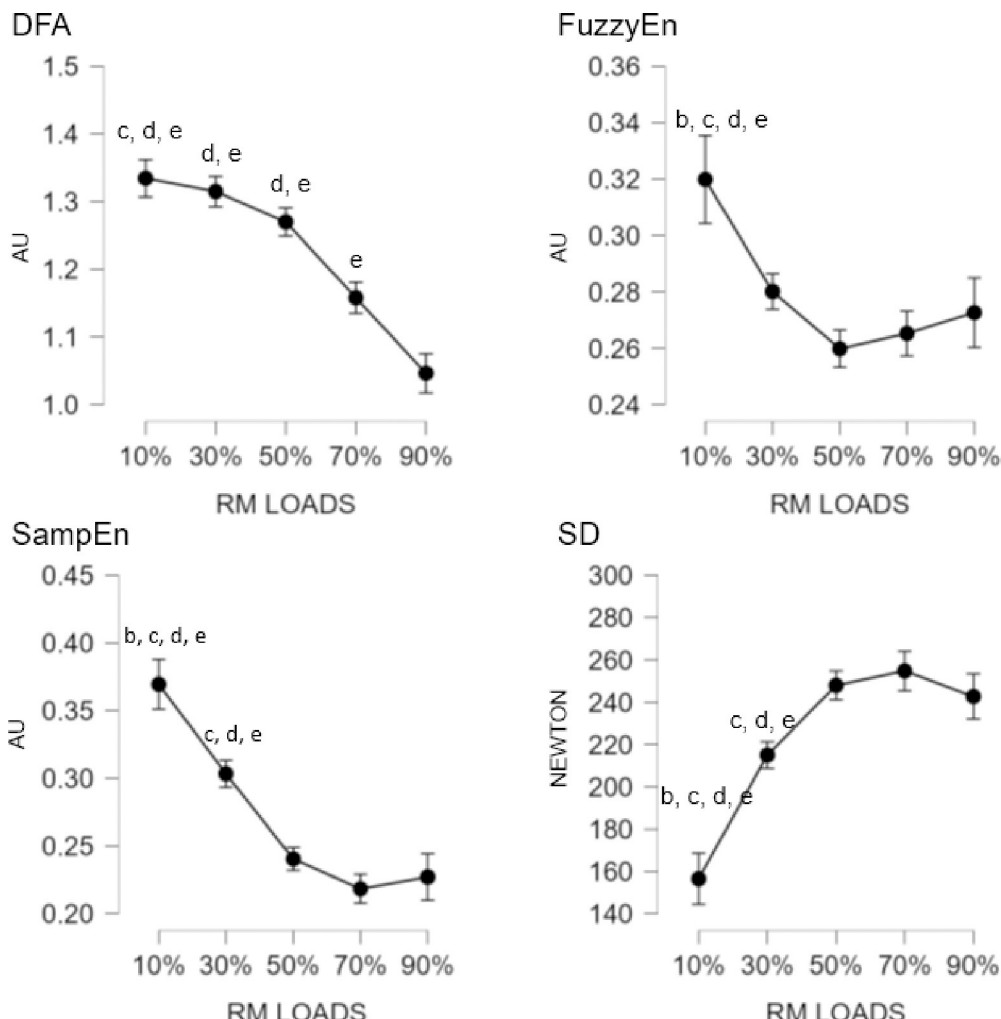

**Fig 1. Mean differences between the different percentages of loads for the measurements with the IMU device.**
AU: arbitrary units. Letter corresponds to the different load comparisons: b = differences versus 30%; c = difference versus 50%; d = difference versus 70%; e = difference versus 90%.

only the DFA variable is sensitive enough to detect changes in force variability when load intensity is modified.

## Load intensity and amount of variability

Increasing load intensity is one variable that can condition motor patterns in squat execution [23]. In our study, the standard deviation showed a significant decrease in the variability of movement with increasing intensity from the 70% load when analyzing the IMU, while the force platform shows an increase in this variable at the first loads, stabilizing from 50%. Our results on the force platform for variability are consistent with those found in other studies where increasing force production increases SD [24–26]. Nevertheless, IMUs showed the opposite trend, with lower SD values of movement acceleration in the higher intensity loads. To understand the reason for these differences, we need to examine the data recorded by these instruments. The force platform records peak forces that are directly influenced by increasing mass with increasing load intensity, as adding weight to the barbell produces higher impulses

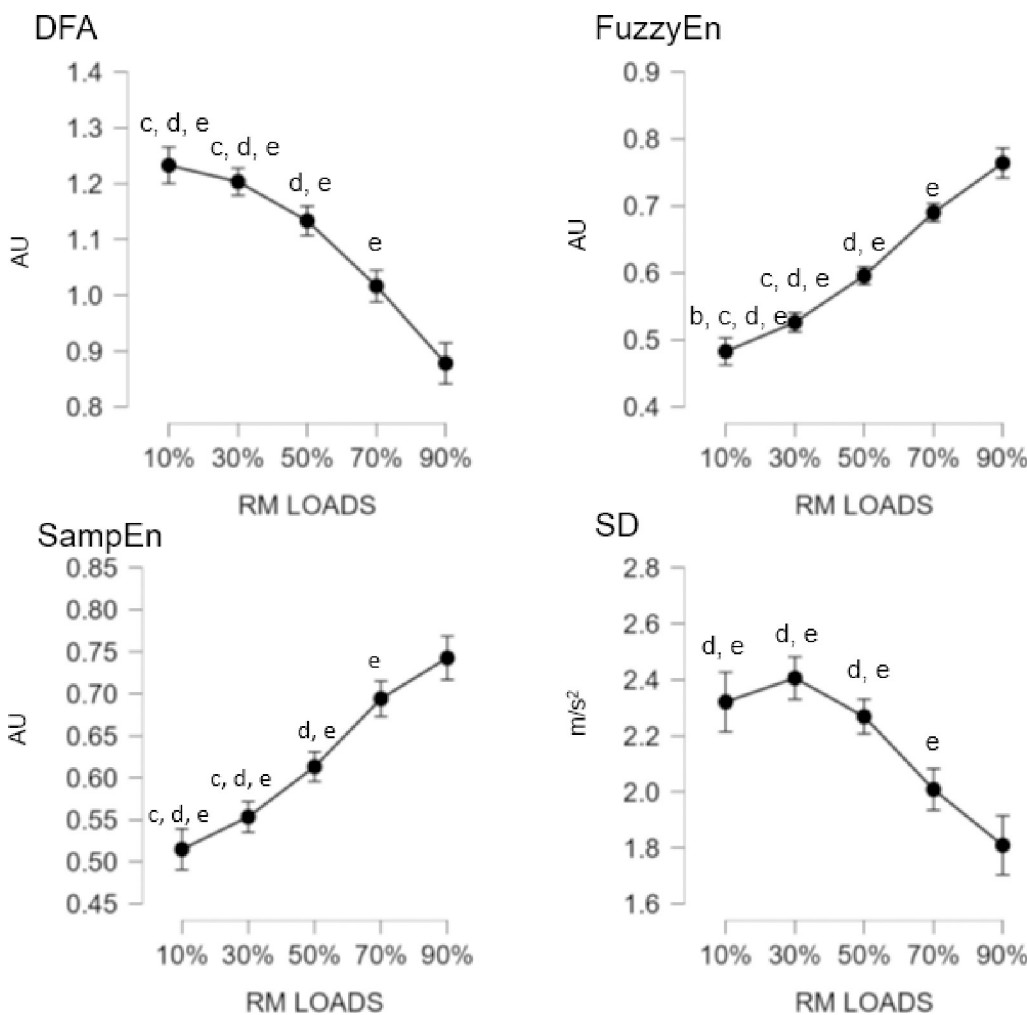

**Fig 2. Mean differences between the different percentages of loads for the measurements with the force platform.**
AU: arbitrary units. Letter corresponds to the different load comparisons: b = differences versus 30%; c = versus 50%; d = versus 70%; e = difference versus 90%.

and peak forces compared to lower loads performing the same movement [27]. Nevertheless, this variability measured by the standard deviation during movement is similar in higher loads, where variations in peak force become less pronounced [28]. On the other hand, IMUs record accelerations, i.e., changes in motion without the influence of mass, and changes in mass do not directly affect acceleration signal as force measured from platform. This indicates that IMUs better reflect adjustments in motion expressed in force variability. Therefore, decreases in SD in our study are similar to other studies looking at other variables, such as execution variability velocity, in which a decrease in velocity variability obtained through IMUs have been observed [29]. In any case, our results show that a linear measure such as SD is insufficient to discriminate load intensity and needs to be complemented by non-linear measures such as motion complexity analysis [6, 7].

## Load intensity and complexity

One of the most important aspects we analyzed in our study is the influence of constraint, how load intensity is a constraint affecting movement complexity, not only on the amount of

movement variability. The increase in load intensity modified the IMUs in different ways and the force plate signal. Thus, complexity from the IMU signal augmented higher loads (higher SampEn and FuzzyEn values, and lower DFA values). Different results were found for the force platform where FuzzyEn, SampEn, and DFA results were lower when load was increased.

The different trends found in the FuzzyEn and SampEn results for the IMU and force platform may be due to what was discussed in the previous section, where the platform records force production in the base of support. These force values are affected by the mass being moved during the squat movement, whereas the IMU only records the acceleration of the movement despite the mass displacement. In addition, the differences found between the loads for FuzzyEn and SampEn of the force platform are at low loads, as is the case for the force platform when measuring the SD. Due to the mathematical formulation of FuzzyEn and SampEn, which includes SD for the calculation of the formula, the range of the signal may affect the resulting FuzzyEn and SampEn values [9, 20]. As the SD was different in the force platform compared to the IMU due to the increase in peak forces caused by the influence of the increased mass, FuzzyEn and SampEn values could show different trends in force platform compared to the IMU. It is precisely these changes in trend and the significant difference that are found with increases in load intensity up to approximately 50% such as in the SD values. In contrast, Fuzzy and SampEn calculated from the IMU signal appear useful for discriminating between loads during squat movement. In opposition to other constraints such as injury [30] or acute fatigue [16], where a reduction in motor variability has been observed (lower entropy values), our results show an increase in motor variability (greater entropy values) according to load increase. Thus, a constraint such as load increases the degree of motor variability to allow an adaptive response to a more challenging situation [14].

These increase in complexity in IMU with increasing load are from the DFA results from the IMU and force platform. The DFA analysis from both devices showed a trend of decreasing DFA values with higher loads, especially up to 30% RM. These results are in accordance with the hypothesis that a more difficult task modifies action in the central nervous system, resulting in greater motor variability for better adaptation to the constraint [31]. Our results in motor variability are different from previous studies in force signal, where submaximal contractions and isometric actions have been preferentially studied [32]. In submaximal actions, the goal of the task can be accomplished with the activation of fewer motor units and the interaction of a reduced number of muscle fibers. Thus, when performing a given task that requires little effort from the system, the reduced number of muscle fibers and motor units needed to complete the task are related to fewer possible solutions of muscle activation, enough to complete the task. In the present study, dynamic and high intensity tasks were selected for the experiment, requiring higher motor unit recruitment, demanding higher intramuscular coordination [33] and, therefore, increasing the complexity of the force output.

From our results, when the difficulty of a task is increased, it becomes more demanding for a person, and movement complexity tends to increase because of adapting to this level of difficulty to successfully complete the task. To the best of our knowledge, no prior studies examined the effect that the load in dynamic strength tasks may have on the complexity of force variability. Previous studies have found increases in variability complexity increasing the difficulty of a strength movement including, for example, an external constraint, such as handling a rugby ball [14], or in dual task conditions in postural control tasks [34]. Other studies on non-fatigued muscles have shown that decreases in variability (lower coefficient of variation, CV) but increases in complexity (higher SampEn) in knee extensor strength are associated with increased performance in balance testing [35]. This increase in balance test performance may resemble the increase in squat performance in our study.

A limitation of our study is that there is a difference in the duration of the time series. This was due to the inherent variability of each individual and the differences in the loads. In addition, in the 90% 1RM series. Some participants were not always able to complete the prescribed four repetitions of this set due to the level of effort required. This may have affected the non-linear tools and the availability of data for inclusion in our analyses. This limitation may provide valuable information for future research and suggest areas for improvement in experimental design. It should also be noted that since there are hardly any studies on acceleration signals in force training, different methodological issues should be investigated in this respect, such as the effect of non-stationarity.

## Conclusion

The introduction of a constraint, such as load intensity, leads to changes in the complexity of the participants' movements. The records of IMUs using non-linear measurements (DFA, FuzzyEn, SampEn) indicate an increase in motion complexity with an elevated percentage load (above 30%). However, at low loads (below 30%) in the IMU, no differences are found (although there is the same tendency to increase complexity), as it is more difficult to find differences when the intensity of the task is less challenging for the system. In the records obtained with a force platform, the complexity is observed to decrease, with changes between loads detected by DFA analysis. These results indicate a decrease in movement variability in terms of amount of variability, while increasing complexity in the IMU device, indicating that as the intensity of the load increases, the system is more compromised and increases its complexity to overcome the task. Therefore, the use of non-linear measures that analyse the complexity of movement through acceleration can be useful for coaches to know the internal load that the different load intensities used on their athletes place on the system. In this way, they will also be able to know the state in which the athlete measuring complexity with a same load in different days or sessions. On the other hand, DFA is a sensitive tool for detecting changes in complexity with increasing load intensity through IMUs and force plates, while FuzzyEn and SampEn are sensitive through IMUs, so all these tools are useful for training practice

## Acknowledgments

We would like to thank A. Oliver-López for his help in recruiting participants for our study. We would also like to thank D. Barbado for his helpful discussions on these results.

## Author Contributions

**Conceptualization:** Francisco J. Moreno, Rafael Sabido.

**Data curation:** Miguel López-Fernández, Fernando García-Aguilar.

**Formal analysis:** Miguel López-Fernández.

**Investigation:** Miguel López-Fernández, Fernando García-Aguilar, Pablo Asencio.

**Methodology:** Carla Caballero, Francisco J. Moreno, Rafael Sabido.

**Supervision:** Francisco J. Moreno, Rafael Sabido.

**Writing – original draft:** Miguel López-Fernández.

**Writing – review & editing:** Miguel López-Fernández, Francisco J. Moreno, Rafael Sabido.

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
