## [Decision Letter · Decision Letter 0]

18 Jun 2024

PONE-D-24-09826Motor variability during resistance training: acceleration signal as intensity indicatorPLOS ONE

Dear Dr. García Aguilar,

Thank you for submitting your manuscript to PLOS ONE. After careful consideration, we feel that it has merit but does not fully meet PLOS ONE’s publication criteria as it currently stands. Therefore, we invite you to submit a revised version of the manuscript that addresses the points raised during the review process.

Hello dear authors,Two reviewers have positively assessed your manuscript. One of them is asking for changes in the conclusion section and to indicate practical utility of your paper. Please make sure to address all comments.  

We look forward to receiving your revised manuscript.

Kind regards,

Danica Janicijevic, Ph.D

Academic Editor

PLOS ONE

“This manuscript is related to a national project funded by the Ministerio de Ciencia e Innovación [PID2019-109632RB-I00]. The contribution of Miguel López-Fernández was funded by the Generalitat Valenciana, Spain [grant number: CIACIF/2021/452]. The contribution of Fernando García-Aguilar was supported by the Generalitat Valencia, Spain [grant number: ACIF/2021/159]. The contribution of Pablo Asencio was supported by the Ministerio de Ciencia e Innovación [PRE2020-091858]. These grants have made it possible to carry out this research.”

“We would like to thank A. Oliver-López for his help in recruiting participants for our study. We would also like to thank D. Barbado for his helpful discussions on these results.

This manuscript is related to a national project funded by the Ministerio de Ciencia e Innovación [PID2019-109632RB-I00]. The contribution of Miguel López-Fernández was funded by the Generalitat Valenciana, Spain [grant number: CIACIF/2021/452]. The contribution of Fernando García-Aguilar was supported by the Generalitat Valencia, Spain [grant number: ACIF/2021/159]. The contribution of Pablo Asencio was supported by the Ministerio de Ciencia e Innovación [PRE2020-091858]. These grants have made it possible to carry out this research.”

“This manuscript is related to a national project funded by the Ministerio de Ciencia e Innovación [PID2019-109632RB-I00]. The contribution of Miguel López-Fernández was funded by the Generalitat Valenciana, Spain [grant number: CIACIF/2021/452]. The contribution of Fernando García-Aguilar was supported by the Generalitat Valencia, Spain [grant number: ACIF/2021/159]. The contribution of Pablo Asencio was supported by the Ministerio de Ciencia e Innovación [PRE2020-091858]. These grants have made it possible to carry out this research.”

Reviewers' comments:

Reviewer's Responses to Questions

**Comments to the Author**

1. Is the manuscript technically sound, and do the data support the conclusions?

Reviewer #1: Yes

Reviewer #2: Yes

2. Has the statistical analysis been performed appropriately and rigorously? 

Reviewer #1: Yes

Reviewer #2: Yes

3. Have the authors made all data underlying the findings in their manuscript fully available?

Reviewer #1: Yes

Reviewer #2: Yes

4. Is the manuscript presented in an intelligible fashion and written in standard English?

Reviewer #1: Yes

Reviewer #2: Yes

5. Review Comments to the Author

Reviewer #1: The research presented by the authors is very interesting and important for the proper motor preparation of players. The authors fulfilled the criteria for scientific works. In the reviewer's opinion, greater care should be taken in the selection of current literature in the future. The authors rightly pointed out the limitations of the research conducted, which will translate into better performance in subsequent studies.

Reviewer #2: It is understood that these four methods are important analyses for accurately analyzing variability in load and intensity during strength training. However, for their application to be more understandable and usable, it is necessary to highlight the loads where variability is detected and not detected in the conclusion section. Additionally, the measures to be taken during the application (for example, the lack of difference between 10% and 30% in DFA) should be addressed. In short, how can coaches or athletes shape their strength training practices after reading this paper? This topic needs to be emphasized.

In this sense, what should coaches and athletes do with the data provided by these four methods regarding strength variability? Should they train at certain 1RM percentages with more units, or is it unnecessary to spend too much time at certain 1RM percentages? For quick adaptation, what should be considered in light of these four methods?

6. PLOS authors have the option to publish the peer review history of their article (what does this mean?). If published, this will include your full peer review and any attached files.

Reviewer #1: No

Reviewer #2: No

---

## [Author Response · Author response to Decision Letter 0]

3 Jul 2024

Authors’ response (AR): We would like to thank the reviewers for their helpful advice and suggestions regarding this manuscript. We found their criticisms and recommendations very constructive. We appreciate the work done in revising this manuscript, and we have responded to your concerns point by point. In the new version of the manuscript, changes have been marked in red.

Reviewer 1

The research presented by the authors is very interesting and important for the proper motor preparation of players. The authors fulfilled the criteria for scientific works. In the reviewer's opinion, greater care should be taken in the selection of current literature in the future. The authors rightly pointed out the limitations of the research conducted, which will translate into better performance in subsequent studies.

AR: Thank you very much for your kind words and for your appreciation of our work. We are pleased to know that you find our study relevant and important for the scientific field and its applicability. We appreciate your support and are grateful for your positive comments.

Reviewer 2

1. It is understood that these four methods are important analyses for accurately analyzing variability in load and intensity during strength training. However, for their application to be more understandable and usable, it is necessary to highlight the loads where variability is detected and not detected in the conclusion section. Additionally, the measures to be taken during the application (for example, the lack of difference between 10% and 30% in DFA) should be addressed. In short, how can coaches or athletes shape their strength training practices after reading this paper? This topic needs to be emphasized.

AR: Thank you for this suggestion. We have added your suggestion in the conclusions section in line 277 indicating “The records of IMUs using non-linear measurements (DFA, FuzzyEn, SampEn) indicate an increase in motion complexity with an elevated percentage load (above 30%). However, at low loads (below 30%) in the IMU, no differences are found (although there is the same tendency to increase complexity), as it is more difficult to find differences when the intensity of the task is less challenging for the system. In the records obtained with a force platform, the complexity is observed to decrease, with changes between loads detected by DFA analysis”. 

2. How can coaches or athletes shape their strength training practices after reading this paper? This topic needs to be emphasized. In this sense, what should coaches and athletes do with the data provided by these four methods regarding strength variability? Should they train at certain 1RM percentages with more units, or is it unnecessary to spend too much time at certain 1RM percentages? For quick adaptation, what should be considered in light of these four methods?

AR: Thank you for this suggestion. In line 283, we have deleted "It appears" and added "These results indicate". In the same paragraph, but in line 284, we have added the following: “indicating that as the intensity of the load increases, the system is more compromised and increases its complexity to overcome the task. Therefore, the use of non-linear measures that analyze the complexity of movement through acceleration can be useful for coaches to know the internal load that the different load intensities used on their athletes place on the system. In this way, they will also be able to know the state in which the athlete measuring complexity with a same load in different days or sessions” to answer your suggestions on the applications of these results for coaches and practical applications. We have also changed the sentence from line 291 onwards and have added “while FuzzyEn and SampEn are sensitive through IMUs, so all these tools are useful for training practice”.

---

## [Decision Letter · Decision Letter 1]

16 Jul 2024

Motor variability during resistance training: acceleration signal as intensity indicator

PONE-D-24-09826R1

Dear Dr. Fernando García Aguilar,

We’re pleased to inform you that your manuscript has been judged scientifically suitable for publication and will be formally accepted for publication once it meets all outstanding technical requirements.

Kind regards,

Danica Janicijevic, Ph.D

Academic Editor

PLOS ONE

Additional Editor Comments (optional):

Reviewers' comments:

Reviewer's Responses to Questions

**Comments to the Author**

1. If the authors have adequately addressed your comments raised in a previous round of review and you feel that this manuscript is now acceptable for publication, you may indicate that here to bypass the “Comments to the Author” section, enter your conflict of interest statement in the “Confidential to Editor” section, and submit your "Accept" recommendation.

Reviewer #2: All comments have been addressed

2. Is the manuscript technically sound, and do the data support the conclusions?

Reviewer #2: Yes

3. Has the statistical analysis been performed appropriately and rigorously? 

Reviewer #2: Yes

4. Have the authors made all data underlying the findings in their manuscript fully available?

Reviewer #2: Yes

5. Is the manuscript presented in an intelligible fashion and written in standard English?

Reviewer #2: Yes

6. Review Comments to the Author

Reviewer #2: (No Response)

7. PLOS authors have the option to publish the peer review history of their article (what does this mean?). If published, this will include your full peer review and any attached files.

Reviewer #2: No

---

## [Editor Report · Acceptance letter]

10 Sep 2024

PONE-D-24-09826R1 

PLOS ONE

Dear Dr. García Aguilar, 

I'm pleased to inform you that your manuscript has been deemed suitable for publication in PLOS ONE. Congratulations! Your manuscript is now being handed over to our production team.

Kind regards, 

on behalf of

Dr. Danica Janicijevic 

Academic Editor

PLOS ONE